# The Human Rights Situation of Intersex People: An Analysis of Europe and Latin America

**Yessica Mestre** [1,2]

1  Andalusian School of Public Health, 18011 Granada, Spain; yessica.mestre.easp@juntadeandalucia.es
2  Doctoral School in Humanities, Social Sciences and Law, University of Granada, 18071 Granada, Spain

**Abstract:** Today, intersex people of Europe and Latin America are subjected to different degrees of invisibility and discrimination for being part of bodily diverse communities. Thus, these experiences have been addressed as violations of intersex human rights. This research intends to explore intersex human rights across Europe and Latin America through a scoping review and legal research, including the review of activist documents. It seeks to study the experiences of violence suffered by intersex people, the role of states in promoting justice, and the possibilities offered by a human rights framework to guarantee a dignified life for bodily diverse communities. This research also illustrates that, although regional, cultural and social landscapes differ in both continents, intersex groups are subject to constant violations of their human rights, and they struggle for recognition and for their bodies to be respected outside the binary categories of sex and gender. Furthermore, European and Latin American states have made proven legislative advances that have led to the greater visibility of intersex people but also face remaining difficulties and gaps.

**Keywords:** intersex; human rights; children; Europe; Latin America





## 1. Introduction

*Intersexuality* is a term that designates various conditions in which a person is born with diverse sex characteristics that differ from traditional definitions of male and female. According to Khanna (2021, p. 186), "to be intersex means that a person has genitals, reproductive organs, secondary sex characteristics, hormones and chromosomes that fall outside the commonly known binary definitions of either male or female sex". Since intersex individuals have characteristics that do not fit into masculine and feminine roles, beliefs and practices, they suffer discrimination and stigmatisation from birth. According to Human Rights Watch, intersex people are subjected to medical treatments that can permanently harm them from infancy (Tamar-Mattis and Knight 2017). In general terms, violence against intersex individuals results from misinformation, stereotypes and myths circulating about intersex. Subsequently, the trauma arising from these acts connects intersex people's conditions with human rights research and practice (Lev 2006).

When addressing the violence committed against intersex people, it is essential to differentiate the concepts of sexual orientation, gender identity, and sex characteristics to distinguish their fight for the respect of their rights from other population groups that may have also suffered discrimination within these terms. The Yogyakarta Principles (2007) refer to sexual orientation as "each person's capacity for profound emotional, affectional and sexual attraction to, and intimate and sexual relations with, individuals of a different gender or the same gender or more than one gender" (p. 6). In addition, the same document conceives the concept of gender identity as "each person's deeply felt internal and individual experience of gender, which may or may not correspond with the sex assigned at birth, including the personal sense of the body (which may involve, if freely chosen, modification of bodily appearance or function by medical, surgical or other means) and other expressions of gender, including dress, speech and mannerisms" (The Yogyakarta

Principles 2007, p. 6). In contrast to these concepts, sexual characteristics can be regarded as "each person's physical features relating to sex, including genitalia and other sexual and reproductive anatomy, chromosomes, hormones, and secondary physical features emerging from puberty" (YP+10 2017, p. 6). Intersex activists emphasise that, by subsuming the notion of sex characteristics within the LGBT human rights framework, there is a risk of creating misconceptions around bodily diverse populations and neglecting important issues that intersex people face, such as epistemic injustice (Carpenter 2015).

In this sense, according to the United Nations "between 0.05% and 1.7% of the population is born with intersex traits" (UN 2017, p. 1), which means that they have variations in their sex characteristics that do not fit the medical and social definitions of male and female. Hence, they are subject to discrimination and other human rights violations based on the variation of their sexual characteristics (Ghattas 2019). According to a previous study on the situation of intersex people in Europe, perfectly healthy intersex individuals have also experienced prenatal interventions and medical interventions on their bodies (Monro et al. 2019, apud in Ghattas 2019). The documentation analysed indicates that the major struggles intersex people face are related to society's perception of the variation of their sex characteristics. Therefore, from an early age, intersex people are subjected to so-called sex-normalising surgeries, which can be understood as irreversible procedures performed on intersex children to assign a sex that fits the gender role according to social expectations (Tamar-Mattis and Knight 2017). These "[i]nvasive, irreversible and non-emergency medical interventions" have resulted in traumatic experiences that intersex people undergo from an early age (Ghattas 2019, p. 15).

Existing research on human rights law and the human rights of intersex people in Europe and Latin America has addressed the role of the law in protecting the life of bodily diverse people. Garland and Slokenberga (2018) have explored medical practices on intersex children to explain how, in Europe, these practices entail human rights violations, according to the guidelines of the United Nations, the Council of Europe and the European Union values. However, few European countries, including Malta, Portugal, Germany and Iceland, have strictly prohibited medical procedures and treatments of intersex children. In Latin America, legal research frameworks have addressed the human rights of intersex people. For instance, Flores-Manzano and Vázquez-Martínez (2021) demonstrated how the human rights of intersex communities have been violated, stigmatised and made invisible in Ecuador without present regulations having achieved greater inclusion. In Chile, the legal problems associated with intersex status have also been investigated.

Grez (2020) argued that the intersex human rights struggle extends to medical practice and law borders. There remain social norms and beliefs that encourage discrimination and violence toward intersex people, which is why protecting them against violations of their human right to bodily integrity is urgent.

Injustice against intersex people in Latin America and Europe calls for research as to whether human rights are enshrined in the legal provisions. A comparative investigation of these matters would also deepen the analysis of intersex people's human rights on both continents and would help us to better understand why such a framework is needed to improve the lives of intersex people. This research explores the current human rights situation of intersex people in Europe and Latin America through a scoping review and a legal research framework that includes the analysis of activist documents. Accordingly, it seeks to understand the general human rights provisions related to intersex people in these continents by also reviewing intersex activist perspectives on these legal developments. Thus, it aims to understand how European and Latin American states have established measures to protect intersex people's human rights at different levels. In addition, this research interrogates the existing gaps between the parameters provided by international law and their application in the national legislation. Both Europe and Latin America have distinct trajectories of legal development, sociocultural contexts and activism for intersex human rights, and may, therefore, provide different approaches. While in Europe the results of activism have encouraged legislation that promotes the protection of the rights of

intersex people (OII Europe 2022), in Latin America, the recognition of their problems has been first acknowledged in the decisions of the courts, and these sentences have been used accordingly as tools to protect the rights of intersex people and advocate for a legislation that guarantees their rights (Pikramenou 2019). The different legal dynamics of these continents are the reason why they were chosen as cases of study for this research.

This article is divided into four main sections. First, the methodological strategy to analyse the current human rights situation of intersex communities in Latin America and Europe will be described. Second, an analysis will be provided of the human rights violations based on gender expression/identity and sex characteristics. Subsequently, a discussion will take place on the current status of intersex human rights in both regions. The final section will present the concluding remarks of the study.

## 2. Materials and Methods

This research is the result of a scoping review and legal research that includes publications and reports of major intersex NGOs and institutions. In this section, the approaches to data collection and analysis are addressed.

### 2.1. Scoping Review

A scoping review was carried out to analyse the human rights situation of intersex people in Europe and Latin America. Scoping reviews have become a well-known technique for synthesising research evidence in epidemiology and synthesis studies, but also in the context of social sciences (Pham et al. 2014). According to Pham et al. (2014), the primary purpose of this review modality is to identify available research evidence on a given topic.

In this research, the scoping review was conducted to analyse the contemporary development of intersex human rights and their most common violations. Consequently, recent literature on the protection of the human rights of intersex individuals was identified, examining their political demands and the legal and public policy gaps concerning their protection.

The literature review was conducted between 2015 and 2021. To identify the correct content, "intersex" OR "DSD" OR "differences in sex development" OR "disorders of sex development" AND "human rights" were used as keywords. SCOPUS, Web of Science and PubMed databases with texts in English were the sources. In addition, Scielo, a database with a predominance of texts in Spanish and Portuguese, was chosen to provide more coverage of the cultural environment, particularly for Latin America. Across the databases mentioned, 165 articles were found. Subsequently, 21 duplicates were removed, leaving 144 articles. Their titles and abstracts were analysed for relevance to the study topic. After this process, 45 articles were selected for inclusion in the final analysis.

### 2.2. Legal Research Approach and Review of Activist Documents

Although this paper is based on the references found in the scoping review, it focuses more thoroughly on the legal instruments for protecting the human rights of intersex people, since the purpose of the study was not only to explore the current situation of intersex human rights in Europe and Latin America but also to examine their effective application. This research is based on the comparative analysis of the legal context of intersex people's human rights in Europe and Latin America.

To better understand this approach, it is essential to explain the legal doctrine used. Van Hoecke described the legal doctrine as a set of specific actions such as interpreting texts, creating arguments, providing empirical evidence for these arguments, creating axioms and logical suppositions, and assuming positionings (Van Hoecke 2011). In a strict sense, the legal doctrine is a normative and argumentative discipline because it deals with interpreting texts and arguments (Van Hoecke 2011). Subsequently, the legal doctrine has been defined as an empirical discipline because it presents research questions that lead to creating statements that are then tested through empirical data collection (Van Hoecke 2011). Furthermore, since legal doctrine is a discipline that explains why norms are valid in

each society, it provides explanations, axioms and logical statements (Van Hoecke 2011). Finally, the legal doctrine is a normative discipline because it describes the norms and systematises them, assumes positions, and adopts values and interests (Van Hoecke 2011).

According to Van Hoecke (2011), the different definitions of legal doctrine led to the adoption of plural methodologies to address the place of norms in a given society (Van Hoecke 2011). As a result, numerous sources can be considered in methodologies to explore legal doctrines. For instance, a researcher can incorporate law cases, scholarly writings and primary sources to create and sustain arguments and normative positions (Van Hoecke 2011). Hence, legal research is an approach that focuses on laws and regulations and the functioning of organisations and institutions in legal research society (Van Hoecke 2011). Leeuw and Hans (2016) also explained that legal research considers the consequences of laws for individuals, organisations and social actors. Thus, the legal research approach assumed in this research included examining publications and reports of major intersex NGOs and institutions. Through the review of these documents, the way intersex activism evaluates the implementation of these laws and regulations in everyday life was analysed.

### 2.3. Data Collection, Systematising and Review

The process of collecting, systematising and reviewing information comprehends different levels. First, the legal issue addressed in this research was described, which is the protection of the human rights of intersex people. In that sense, numerous sources were considered, such as jurisprudence of national and international courts, recommendations of international organizations, statements and publications from intersex NGOs, with 43 legal documents and 27 publications of intersex NGOs reviewed. Additionally, attention was paid to the institutions, and a distinction was made according to their *international*, *regional* or *national* scope. The institutions analysed in this research were the United Nations, the European Parliament, the Council of Europe, the European Court of Human Rights, and the Inter-American Commission on Human Rights. Regarding the national legislations, the countries of Spain, Belgium, Germany, Iceland, Portugal, Malta, Colombia, Mexico, Argentina, Chile and Costa Rica were studied to establish legal comparisons. Finally, this research collected experiences and data from declarations and reports made by the NGOs OII Europe, ILGA-Europe, Intersex Human Rights Australia–IHRA, Brujula Intersexual, STOPIGM, and other individual intersex activists.

### 3. Results and Discussion

### 3.1. Human Rights Violations Based on Gender Expression/Identity and Sex Characteristics

Traditionally, the presence of differences within a diverse society, such as the contrasting ways of expressing one's sex, gender identity or the diversity of bodies, has promoted discriminatory scenarios where a person becomes a subject of violence or marginalisation because of their bodily difference (UN 2001). Discussing these scenarios exposes the legal systems' difficulties in eradicating unequal treatment and discriminatory practices. The scoping review and legal research, including the review of activist documents, explained a number of essential advances in intersex human rights, elucidating experiences and mechanisms for their protection. It is essential to highlight how intersex organisations assert that gender recognition is not enough to protect the right to the physical and mental integrity of intersex people (IHRA 2019). In addition, social movements have highlighted how significant it is not to confuse sex and gender when considering intersex human rights, which are mainly determined by sex characteristics and embodied experiences (IHRA 2019).

These legal developments, debates and disputes can be interpreted in the context of the Convention on the Rights of the Child (CRC) (UN 1989) and the United Nations Convention against Torture and Other Cruel and Inhuman or Degrading Treatment or Punishment (UNCAT) (UN 1987) to elucidate the struggles against cruel medical procedures, self-recognition and identity, and the right to truth. There are differences in how these legal documents mention intersex human rights. On the one hand, the Yogyakarta Principles

plus 10 (YP+10 2017) explicitly address the human rights of intersex people. On the other hand, the CRC (UN 1989) and the UNCAT (UN 1987) affirm the universal application of human rights that can be used for intersex individuals.

The Yogyakarta Principles (2007) were launched in 2007 by experts and activists from different fields to apply the Universal Declaration of Human Rights (UDHR) (UN 1948) in response to massive human rights abuses related to sexual orientation, gender expression/identity and sex characteristics. The initial idea of this civil society manifestation was to articulate universal values applicable to sexual orientation and gender identity and to clarify concepts to avoid discriminatory treatment (O'Flaherty 2015). Since its creation, this statement has achieved an international influence and positive impact; however, there were issues that the first version did not explicitly cover, and, in 2017, additional principles were developed to extend some topics. The guarantee of the protection of human rights of intersex people was one of the main aims of the new approach. Thus, the YP+10 (2017) considers sex characteristics and relevant situations for intersex people.

There are serious human rights violations associated with bodily perception. In consequence, the YP+10 (2017) includes crucial matters, such as the right to bodily and mental integrity, the right to legal gender recognition, and the right to truth. In addition, the YP+10 (2017) proposes states' obligations that are essential for the legal defence of intersex human rights. The YP+10 Principle 32, which mandates the right to bodily and mental integrity, and its obligation "D" prescribe how intersex child protection must be developed:

STATES SHALL: ( . . . )

(D) Bearing in mind the child's right to life, non-discrimination, the best interests of the child, and respect for the child's views, ensure that children are fully consulted and informed regarding any modifications to their sex characteristics necessary to avoid or remedy proven, serious physical harm, and ensure that any such modifications are consented to by the child concerned in a manner consistent with the child's evolving capacity. (YP+10 2017, p. 10)

This obligation focuses on one of the main aspects of the defence of intersex human rights, the agency of children and their best interest when performing medical procedures at an early age (YP+10 2017). Carrying out medical interventions which are strongly influenced by social stereotypes and not biological needs is a violation of the UDHR (UN 1948). Article 3, which concerns the right to life, liberty, and security of the person, and Article 5, linked to the prohibition of torture, punishments, or degrading treatments, can be interpreted in relation to the protection of intersex children against abuse presented by medical treatment (UN 1948). The same tenet is also established by Article 7 of the International Covenant on Civil and Political Rights (CCPR) (UN 1966), which prohibits torture or degrading conduct against people, specifying that a human being cannot be subjected to experimentation without her/his/their free consent. In addition, the CRC (UN 1989), in its Article 24, determined that States Parties must implement all effective measures to abolish traditional health practices, which also applies to the protection of intersex children. Consequently, Article 37 prescribes that no child shall suffer any type of degrading treatment or punishment.

Concerning the right to legal recognition, the YP+10 (2017), in its Principle 31 and obligation "B", points out the state's obligation to ensure modification of names, based on the self-determination of the person: (B) Ensure access to a quick, transparent and accessible mechanism to change names, including to gender-neutral names, based on the self-determination of the person. (YP+10 2017, p. 9)

Administrative procedures linked to gender assignment at the moment of birth and the desire to change this in the future should not be discriminatory and, whether or not they actually are discriminatory, they should be considered a violation of the UDHR (UN 1948). Article 6, which mentions that "everyone has the right to recognition everywhere as a person before the law", represents this mandate. Moreover, Article 7 affirms that all people are entitled to equal protection before the law without discrimination. In the same vein, the CCPR (UN 1966) in Article 16 states that "everyone shall have the right to

recognition everywhere as a person before the law". Furthermore, the CRC (UN 1989) in Article 7 recognises "the right to every child to acquire a nationality". Moreover, Article 8 clearly states that when a "child is illegally deprived of some or all of the elements of his or her identity, States Parties shall provide appropriate assistance and protection, with a view to re-establishing speedily his or her identity". Finally, the CRC completed the ruling with Article 8, n. 2, which calls on the states to provide appropriate assistance and protection to re-establish the rights of children when they are "illegally deprived of some or all the elements of their identity". In this sense, these legal tools provide human rights-based arguments for parents of intersex people who want to classify them at birth in a binary category or would like to wait until the intersex person decides to modify the category initially assigned when they are older.

The third important statement for intersex people in the YP+10 (2017) is Principle 37, the right to truth, which contains an important state duty to intersex people in its obligation "C". The YP+10 (2017) directly mandates the protection of intersex people's rights to know the truth about their medical histories. The circumstances surrounding invasive medical interventions related to the bodies of intersex people in the early stages of life are accompanied by secrecy in the procedures, the prohibition of the parents from telling their children about the interventions performed, and the denial of access to medical records (Horowicz 2017). Intersex people are denied the truth about the medical procedures performed on their bodies, and how these procedures can impact their body and identity, and this constitutes a violation of the right to autonomy, identity and personality.

It is possible to appreciate that the legal framework for claiming the right to truth and protecting intersex people against these violations is not as well developed as the right to bodily and mental integrity and its link with torture, or the legal recognition and its link with equal treatment before the law. However, the right to truth has been established under the umbrella of transitional justice, understanding transitional justice as the process of a society from conflict to peace after systematic human rights violations. In this context, the achievement of justice during the restructuring of society gives legitimacy to the new state institutions (Sweeney 2018).

There are intersex human rights violations related to medical procedures, self-recognition and identity, and truth. The scoping review and the legal research framework with the review of activist documents illustrated how the knowledge produced on the human rights of intersex people represents a predominant interest in the first two legal matters. The right of intersex communities to the truth has been tenuously addressed in legal research because this issue has not been fully developed in international human rights law, either. Consequently, it is important to inquire about the collective struggle of intersex persons for their right to the truth, which includes complete knowledge about the medical procedures performed on their bodies.

### 3.2. Current Human Rights Situation of Intersex People in Europe and Latin America

The protection of intersex people's human rights has been a matter of recognition and awareness in Europe and Latin America, acquiring more visibility in the last decade thanks to intersex advocacy and activism efforts. Although there have been significant steps toward constructing intersex human rights frameworks, there is still a gap between the parameters provided by international law, which is not strictly enforceable, and the application of these in the national legislation.

### 3.2.1. Europe

The research included a review of resolutions, recommendations and reports of the European Union institutions, as well as intersex-related legislation in some countries of the European continent were analysed. The European Union (EU) has an institutionalised structure distinguished by its strong representation in fields such as human rights protection. Its legal framework is characterised by fundamental values that are not subject to exceptions, with a horizontal system for the protection of fundamental rights, which means

the prevalence of these rights must be considered by all internal bodies, regardless of the topics they cover (European Parliament 2021). Moreover, there are specialised bodies that deal with the topics directly related to human rights protection: the Council of Europe (CoE), the European Parliament, the European Union Agency for Fundamental Rights (FRA) and the European Court of Human Rights (ECHR).

Over the past few years, EU bodies have issued resolutions and recommendations for its member states to stop the practice of sex-normalising surgeries on intersex children. In 2013, the Parliamentary Assembly of the Council of Europe (PACE) was the first to show its concern for the physical integrity of intersex children regarding the performance of cosmetic surgeries within the EU (PACE 2013). Years later, the FRA (2015) and the Commissioner of Human Rights of the Council of Europe (CoE 2015) issued reports on the situation of intersex people and the need to protect their human rights in Europe. Furthermore, in a 2017 resolution, the PACE (2017) confirmed that these surgeries have shown no " . . . evidence to support the long-term success of such treatments, no immediate danger to health and no genuine therapeutic purpose for the treatment" (p. 1). Following this idea, the European Parliament (2019) pointed out in resolution in 2019 the "urgent need to address violations of the human rights of intersex people, and calls on the Commission and the Member States to propose legislation to address these issues" (p. 4).

European laws and courts' rulings are not only tools to interpret legal principles but also a means to struggle for intersex rights. Some of these legal documents directly mention intersex people, while others assume that the rights of bodily diverse communities are of direct application in the human rights field. Regarding torture, the European Court of Human Rights (EctHR) has extended the scope to consider the applicability of the European Convention of Human Rights, Article 3, when (i) the treatment performed is inhuman and degrading, which includes not only physical but also mental suffering (CoE 2021), and (ii) such treatment reaches a deep level of psychological anguish.[1] In addition to these requirements, the intensifying criterion to classify the actions under Article 3 is "the suspicion that it is committed under the inevitable context of discrimination based on sex, age, race, ethnic origin and religion"[2] or that it is committed "particularly to children and other vulnerable members of society", which requires reinforced protection as a positive obligation of Article 3.[3] Furthermore, concerning the case of medical interventions, the ECHR has emphasised that the procedures that violate physical integrity must have a solid justification for creating such an amount of suffering.[4] Moreover, the states have a positive obligation to prevent the commission of ill-treatment and investigate the possible violation of human rights that covers itself with the redress of the damage, according to Article 13 of the Convention.[5]

As reported in the Council of Europe's issue paper, for intersex children, the medical procedures performed, which begin at birth and in childhood, and later hormonal treatments in adolescence and adulthood may lead to lifelong physical and mental consequences (CoE 2015). In addition, the intensity of an unnecessary clinical intervention can cause bodily humiliation, great pain and traumatisation; thus, the report describes how medical diagnoses given to an intersex child become a reason to transform their body without their consent and provoke unnecessary suffering. Thus, the Council of Europe (CoE 2015) expressed disapproval of these corrective surgeries and stated the need for the legal prohibition of these practices.

Establishment of positive obligations regarding these practices has been achieved by some European countries that have become leaders in the protection of the human rights of intersex people. Malta, Portugal, Iceland and Germany have achieved greater protection of the fundamental rights of intersex people, particularly children, who are the most vulnerable to unnecessary medical interventions during the first stages of their lives. Meanwhile, Belgium and Spain are currently in the legal process of creating their national regulations regarding intersex human rights.

Malta was the first country to recognise and protect the rights of a person based on their sex characteristics. Its government enacted in 2015 a law called the Gender Identity, Gender

Expression and Sex Characteristics Act,[6] which, besides ensuring the self-determination and expression of gender identity for Maltese citizens, condemns any medical treatment or intervention performed to adjust children's sex characteristics without their informed consent. Furthermore, the law contemplates the parents' consent as a replacement only under exceptional circumstances and the support of an interdisciplinary team elected by the government to ensure the child's best interests.[7] According to intersex activism, this law is vital because it protects the territory of sex characteristics and mandates caring for the children's physical integrity (ILGA 2015). Additionally, the law forbids genital mutilation, which is one of the most disturbing concerns for intersex people.[8]

Along the same lines, Portugal adopted a law in 2018 to regulate the same two concerns: the self-determination and expression of gender identity and the protection of the physical integrity of an intersex person against modifying their sexual characteristics. This Portuguese law prohibits medical interventions on intersex people "until the moment that is a manifestation of their gender identity" (Article 5) and as long as there is no "proven risk" to the health of the intersex person.[9]

Another notable legislative act is the one issued by the government of Iceland in 2019[10] which focuses on gender recognition. The law regulates important aspects regarding intersex people's rights in its Article 11a, according to which: (i) permanent changes "shall only be made in conformity with the will of the child and its level of gender identity, and always with the best interests of the child in mind"; (ii) the duty of hospitals is to provide "counselling and support" for the guardians and child; (iii) the possibility for the guardians and child to "seek expert opinion outside the team on the necessity of such treatment, free of charge" is required; and (iv) an important step for data collection is the mandate that medical institutions "enter information on the treatment into the health record and provide the Director of Health with information on the number and nature of surgical operations and medication and the age of those who undergo these changes." This law considers many crucial aspects for the protection of the rights of intersex people. Sadly, regarding the right to bodily integrity, there are some exceptions. The law excludes hypospadias and micropenis from these regulations, indicating that in these cases the permanent changes "shall not be performed unless a detailed assessment of possible advantages and consequences in the short and long term has been undertaken, including the consequences of not performing a surgical operation or providing medication or postponing it until the child can express its will." This decision has been made even when there were reports documenting the serious human rights violations that these procedures constitute for intersex people (StopIGM 2020a).

Because the laws from Malta, Portugal and Iceland address similar aims, their effectiveness can be compared. Intersex activists in Europe have pointed out that even though these countries recognise the need to protect intersex people from these medical abuses, there are still extensive gaps in the current legislation (StopIGM 2019a). Regardless of the norm, paediatric units of public and private hospitals still consider intersex traits as "abnormalities", and therefore, they openly acknowledge the need for a surgical "correction", "reconstruction", or "repair" of the minor's body (StopIGM 2019a, p. 56; 2019b, p. 21).

Thus, these three countries have established regulations prohibiting unnecessary medical interventions but they have not defined a clear legal consequence if those prohibitions are carried out. If the treating doctor performs surgery without informed consent and there is no imminent risk to the health of the child, the three countries' laws fail to define the specific type of sanction (administrative, civil, or criminal). If the norm prohibits specific conduct, their legal duty would be to establish the consequences for its non-compliance, but since there are no clear judicial measures, it is impossible to advance to a stage of compensation/reparation for the victims. This results in another serious legal gap in the legislations.

In the cases of Germany and Belgium, both countries recognised the right of intersex people to choose a non-binary or diverse option in the civil register. Regarding the German civil register, there are two regulations, one created in 2013 which prescribes that a child who

cannot be assigned to the male or female sex should be registered without any legal gender, and a later gender register could be added with a medical certificate that proves the binary sex.[11] The second regulation in 2018 gave the option for intersex babies to be registered in a category named "divers".[12] Both regulations were criticised by intersex activists since they potentially increase the pressure for gender assignation and the stigmatisation of a child with a medical diagnosis related to intersex traits (OII Europe 2013, 2018; ILGA Europe 2018). Additionally, the German Congress lately approved a law to ban unnecessary medical procedures for intersex children until they can express their full and informed consent or until it is approved by the Court.[13] However, according to activists, some loopholes would allow these surgeries to be performed; if the intersex condition is not reported, doctors can implement any treatment of the person, and again, there is no concrete legal consequence for violating the law (Haug 2021; Bundesverband 2021; OII Europe 2021a).

In Belgium, the situation is also recent. In 2021, the House of Representatives called on the government through a resolution to legislate in the protection of the physical integrity of intersex children and ensure the prohibition of medical treatments without the informed and full consent of the intersex person.[14] Currently, this bill is in progress, and intersex activists have maintained the importance of this step in protecting the rights of intersex people, emphasising that further legislation is needed to comprehensively cover points that the resolution did not mention regarding harmful practices (OII Europe 2021b).

Lastly, Spain constitutionally recognises the autonomy of its regions, which enjoy a certain level of independence in the administration of some of the national mandates,[15] thereby introducing a diverse range of rights protections. Hence, the legal status of medical interventions depend on the region where the intersex baby is born. For instance, if the birth takes place in Madrid, Aragón, Murcia or the Canary Islands,[16] the regional legislations protect the children and prohibit unnecessary medical procedures to the intersex person. If, on the contrary, the person is born in another region, the legal protection to safeguard the rights to bodily integrity will depend on the legal advances that community has made in strengthening the protection of the human rights of intersex people. Thus, some intersex people in Spain have greater guarantees of legal protection of their rights, whereas others may encounter obstacles to obtaining legal recognition.

Regarding national legislation, Spain does not have a law that directly protects intersex people's human rights. However, a draft bill is in progress for the effective equality of trans people and the guarantee of LGTBI rights.[17] This draft bill has a depathologising approach, where the intersex person will have comprehensive healthcare, and forbids any genital modification practices in the newborn, except "in cases where medical indications require otherwise to protect the health of the person".[18] Another aspect to highlight is the one-year period that the government will give the parents of the intersex baby to register the sex in the civil registry, in the light of this draft bill.[19]

Aside from these three states, most European states lack harmonisation between the decisions issued by European entities and their materialisation in national legislation regarding intersex people's rights. Regardless of the strong recommendations made by the European institutions, according to the FRA report published in 2015, "sex (re)assignment or sex-related surgery seems to be performed on intersex children, and young people, in at least 21 EU Member States" (FRA 2015, p. 6). This means that, despite of the deficiency of evidence proving these so-called corrective procedures to be necessary, most countries continue to neglect the creation of legislation that prohibits non-consensual surgeries on sex characteristics.

In France, according to the Periodic Report on the CCPR elaborated by the NGOs StopIGM and GISS/Alter Corpus, numerous forms of intersex genital mutilation continue to be practised, such as unnecessary medical procedures and cosmetic genital surgeries (StopIGM 2020b) through public institutions such as the Reference Centre for Rare Diseases.[20] The national guidelines contemplate numerous treatments for intersex newborns

without medical necessity, interventions like "'masculinising' genital surgeries, 'feminising' procedures, and sterilising procedures are carried out in public university clinics to intersex children where approximately 86% are under four years" (StopIGM 2020b, p. 20). These procedures are justified as a means, for example, to "restore functional genital anatomy to allow future penetrative intercourse (as a male or a female)", "avoid stigmatization related to atypical anatomy" and "respond to the parents' desire to bring up a child in the best possible conditions" (StopIGM 2016, p. 10).

This same line of action has been taken in other European countries, for example, in the UK no specific rule bans intersex surgeries (Nelson 2018). Therefore, these surgeries can generally be performed with valid legal consent. However, children and infants cannot give informed consent, which means British law assumes the parents' responsibility is to provide informed consent under the Children Act (Nelson 2018). In another 'shadow report' prepared by the Intersex NGOs in the UK, it was argued that intersex genital surgery practices are still widely practised in the UK (StopIGM 2019d). As in France, intersex child surgeries are permitted through the public health system and facilitated by third parties (StopIGM 2019d). Although this country is committed to preventing surgeries and any cruel or degrading treatment against intersex people, medical procedures for bodily diverse children are still being normalised to remove their "atypical" sex traits (Monro et al. 2017, p. 11). Hence, intersex people continue to be victims of serious human rights violations with procedures such as "masculinising genital surgical procedures that are unnecessary for basic functioning are routinely advocated and performed in NHS hospitals for hypospadias, for social rather than medical reasons" or "clitorectomies which are entirely unnecessary for medical reasons" (Monro et al. 2017, p. 11).

Another important human rights violation observed in the research is the lack of compensation and recognition of the intersex victims for the mentioned practices. For instance, in Austria the absence of data collection and monitoring and the difficulties intersex people have accessing their clinical records has created a hostile environment for them to claim their rights in the court (StopIGM 2019e). The consequence of this is a culture of impunity and difficulty of access to justice.

The previous national situations made it possible to appreciate the diversity of legislation and current human rights protection in Europe. The scoping review and the legal research framework with the review of activist documents showed to what extent major violations of human rights prevail in European laws, especially in terms of medical procedures and gender recognition. In this regard, European countries' realities are heterogeneous and, therefore, the degrees of recognition of the rights of intersex people are diverse. There are countries with conservative cultures and restrictive gender recognition legislations that are reluctant to guarantee a dignified life for LGBTI communities and particularly for trans and intersex people, such as Hungary, Lithuania, Slovenia, Sweden and the United Kingdom. On the contrary, there are countries that demonstrate greater inclusion in legislation, which is evident in how intersex people have used the law to defend equality. This situation has particularly prevailed in Malta, Portugal, Germany, and Iceland, and it is in progress for Belgium and Spain. According to the results of this research, none of these countries has completed its development or public discussion of the right to truth as a guarantee of the achievement of social justice for the intersex community.

### 3.2.2. Latin America

Latin America has a history of social movements claiming participatory actions reflecting diversity and experiences that are mirrored in the exercise of rights (Dagnino 2006). Hence, the countries of Latin America share a common past of violence in addition to cultural beliefs which promote sexual hierarchies and endonormativity practices that legitimise violence and punishment of people with sexual orientations, gender expressions/identities and sex characteristics that differ from the norm. Usually, this violence is nurtured by traditional conceptions of men's and women's roles in society. In this way, the protection of human rights and the creation of anti-discrimination laws for the LGBTI

population in Latin America are invisible to state authorities, leaving the region with one of the highest rates of violence against this population (SinViolenciaLGBTI 2019).

The Inter-American Commission of Human Rights (IACHR 2015) has defined the violence in the Latin American region against the LGBTI population as a social phenomenon called bias violence. According to Motta and Sáez (2008), this violence emerges from the preconception that all members of a particular group must have or develop specific characteristics of their community. In this sense, a person's appearance with different bodily expressions or gender identities creates negative attitudes and value judgements (Motta and Sáez 2008).

Regarding the specific concerns of intersex people, regional recognition and awareness have increased considerably in the last decade. For example, in 2013 the IACHR (2013) held the first public hearing to understand intersex issues in the region. Intersex activists and advocates expressed their experiences: the types of violence, not only medical, that intersex people suffer during the process of "normalisation"; the clinical methods of standardising intersex bodies and their irreversible consequences; and the national legislations which directly violate international children rights. As a result, the Commission recognised the severe violations of human rights that intersex people undergo for having a diverse body and gave suggestions to the countries to promote good practices (IACHR 2013).

The IACHR (2015) stressed the severe human rights violations that the states commit, such as forced sterilisation, genital "normalisation", denial of medical records and health insurance services, secrecy in the procedures, and the absence of informed consent. Conclusively, it encouraged the member states to create public policies and legislation to stop unnecessary medical interventions without free and informed consent; to provide support to families and intersex children; to generate awareness campaigns on the effects of "normalisation" interventions of intersex children; and to develop educational campaigns to end the stereotypes and invisibility that surround intersex people (IACHR 2015).

Unlike Europe, Latin America does not have a clear regionalisation process, which means that, although there are regional entities, normative unification processes are difficult to achieve; this leads to a significant differentiation in the legislation and gathering of data for the region, which is also heterogeneous in the legal advances regarding the protection of intersex human rights (Rueda 2009). Nevertheless, forums and collaborations have been conducted to pursue regional activism. For instance, the San José de Costa Rica Statement was signed in the First Latin American Regional Conference of Intersex People in 2018. This conference was sought by intersex activists to generate collective demands in Latin America and the Caribbean for stopping human rights violations by the states, regional institutions, medical constituencies and the media (Declaración de San José de Costa Rica 2018). The second version of this regional activism meeting was held in 2020 in Argentina; around 30 activists gathered to discuss the human rights situation of intersex people in Latin America (Balderrama 2020).

There have been jurisprudential advances in Latin America and the Caribbean; it is possible to find examples of protection of intersex people from abuses of their human rights. Since there is a considerable difference in terms of the development of intersex human rights among the countries, it is challenging to make a general review of the topic in the region; instead, it is more accurate to highlight the major advances of some nations and name specific state regulations whose modification should be considered.

Among the positive precedents in intersex human rights protection, Colombia should be mentioned. In the 1990s,[21] this country was one of the earliest in the world to establish the right to the autonomy and bodily integrity of a person when their sex characteristics differ from the traditional binary models (Zelada and Nicoli 2019). Their Constitutional Court (CCC) developed parameters to consider whether it is possible or not to perform a medical intervention: (i) the urgency of the treatment; (ii) the risk and impact on the current and future autonomy of the minor; and (iii) the age of the minor. These parameters force the examination of the consequences of such interventions in the minor's life, and if the

treatment is invasive and irreversible, the minor must wait until they are old enough to decide for themselves.[22]

For the protection of the rights of intersex people, in 1999 the Court made a judgement on whether the parents could authorise a medical and surgical intervention for readjusting the genitals of an intersex infant.[23] The Court ruled that the intersex minor must decide the gender with which they identify, and hence, the ruling indicated that the age of five years is the minimum threshold for a person to consent in an informed way to the possible invasive treatments. Also, it established that parental consent is legitimate for children under this age only if informed consent is sufficiently capable of efficiently supporting the decision.[24]

However, in Colombia, no law prohibits the authorities from executing unnecessary medical interventions on intersex children. According to the document issued by the Capital City Hall, "Diagnosis of the situation of intersex people in Bogotá" (Alcaldía Mayor de Bogotá 2014, own translation), the problem with intersex invisibility in Colombia and the consequent lack of normative advances is the weak consolidation of intersex as a political identity. Unlike the lesbian, gay, bisexual and trans sectors, in the intersex community there is an absence of leadership that reacts to the confusion and ignorance around being an intersex person (Alcaldía Mayor de Bogotá 2014).

There are other relevant Latin American examples that have improved the legal protection of intersex human rights. For instance, Argentina's evolution in the recognition of gender identities has continuously improved, as was the case with the Gender Identity Law enacted in 2012, which opened the door to (i) creating a gender identity where people can modify their body without third party considerations, ordering the state to cover the necessary surgeries and hormonal treatments as part of the Mandatory Medical Plan, (ii) offering guidelines for human rights-based procedures in the rectification of sex registration, and (iii) recognising the citizen's right to decent treatment, allowing themselves the gender identity they want, including the use of a different name from the one that appears in their legal documents.[25]

This law is a worldwide example because it is not framed in binary categories; on the contrary, it encourages respect for diversity and self-perception of gender and body, which has become a crucial argument for international trans and intersex advocacy (OUT-Right 2012). Regarding exclusively intersex matters, in 2020 Argentina proposed a bill elaborated by intersex and LGBTI activists to guarantee the protection of human rights of people based on their sex characteristics.[26] The bill includes aspects such as the right to bodily and sexual diversity, the prohibition of any procedure without the full and informed consent of the person whose body is involved, the right to receive truthful information about the people's sex characteristics, the right to determine their sex on the medical certificate, the right to be registered or modify the civil register, the right to have proper social security assistance, the protection against acts of discrimination, and the creation of a truth commission to clarify the events that previously occurred in the country related to body modification procedures for people with variations of sex characteristics.[27]

It is expected that the approval of this law will change the current situation in Argentina for intersex people, since genital mutilations and other medical procedures are still being carried out in health institutions that continue to recommend early interventions in babies born with intersex traits (StopIGM 2017, p. 108). In this sense, there remain many Latin American countries that sponsor, through their public health institutions, unnecessary medical procedures and pathologising treatment for intersex people (StopIGM 2017, p. 6). For instance, protocols to quickly frame the newborn intersex child in a specific gender are implemented in Brazilian hospitals. Consequently, medical personnel can proceed with medical intervention, enforcing under a pathologising language the idea of sex assignation on behalf of the parents (Machado 2009).

In Mexico, according to the Periodic Report of the CCPR elaborated by the NGOs StopIGM and Brújula Intersexual (StopIGM 2019c), there is (i) an absence of national regulations to stop cosmetic surgeries in intersex children, (ii) a lack of administrative measures to enforce the application of the little existing guidance for these people in

hospitals, (iii) indiscriminate medicalisation of bodily diversity accompanied by degrading treatment such as repeated genital examinations and photographs for scientific research, which constitute harmful practices, and (iv) pathologising recommendations of early interventions for children in order to "deal" with intersex traits. Thus, the report mentions that these practices remain typical, encouraging public health institutions and paediatric associations to follow treatment guidelines that go against the human rights of intersex people (StopIGM 2019c). These are intended under the umbrella of early intervention to prevent future stigmatisation, create "normal" appearances, avoid cancer risk and reduce possible "negative feelings" for the children and their parents. (StopIGM 2019c).

Since the region represents a significant gap in creating legal protocols, Costa Rica is another example where hospitals implement their procedures to effect medical interventions (Rueda 2017). In their medical institutions, it is more suitable to hide the intersex condition, so the parents will not question the interventions or "create confusion where there is none" (Rueda 2017, s.p.). Thus, intersex traits are still considered a pathology for healthcare providers, and intersex activism urgently demands change (Rivera and Jiménez 2017).

As in Europe, Latin American states lack governmental regulations that strictly forbid genital mutilation and unnecessary procedures for intersex children (Rivera and Jiménez 2017). For instance, Chile issued in December 2015 a legal notification ordering a stop to unnecessary treatments of intersex children (Ministerio de Salud, Gobierno de Chile 2015). Nevertheless, one year later, the government issued a new notification contradicting the initial one and falling back in protecting the rights of intersex children (Ministerio de Salud, Gobierno de Chile 2016). It defined the condition of hypospadia as a medical emergency, implemented the term Disorders of Sex Development, and changed the wording of the text from the "need" to defer to the "possibility" of deferring surgery until the patient can show trends of sexual identity, among other changes (Ministerio de Salud, Gobierno de Chile 2016). For activists, this new regulation was considered a pathologisation of intersex traits and their further medicalisation. Thus, this notification justified procedures that extend from "the intervention through unnecessary and non-consensual hormonal therapies that have repeatedly resulted in serious consequences to the health of the intersex individual, to surgeries that can be compared to forced sterilisation and genital mutilation procedures"[28] (Inter and Aoi 2016, s.p.).

This scoping review and legal research, including the review of activist documents, demonstrated that Latin America has a long history of human rights law developments and struggles. In this context, intersex people have used international human rights law to defend their freedoms and lives against degrading medical treatment, violence and discrimination. Since Latin America is part of a heterogeneous continent and does not display a solid regionalisation process as in Europe, it is essential to analyse what happens in each country regarding the human rights of intersex people, who in some cases have joined LGBT movements to make more visible the failures of the state toward them. Although Latin America continues to experience setbacks in the defence of the rights to sexual and bodily diversity, there are countries that have stood out worldwide, either because their courts have developed the framework to guarantee human rights, as in Colombia, or because they are making extraordinary progress, as in Argentina, or have exhaustively researched intersex issues through strong activism, as in the case of Mexico (Alcántara 2019). However, as in Europe, in Latin America, the right of intersex people to the truth continues to fall into a gap in human rights research.

## 4. Conclusions

This research explored the current status of intersex human rights in Latin America and Europe. To undertake this aim, a scoping review and legal research, including the review of activist documents, were conducted to allow a better understanding of the experiences of intersex individuals in these regions. Because of this methodology, questions about how European and Latin American states have addressed the human rights of intersex people and the gaps in legislative measures to protect them were approached. Although

Europe represents a regionalisation process that translates into parliaments, commissions and courts, Latin America has shown progress in terms of the inclusion of intersex people through the IACHR and national courts. Thus, this research identified existing legal sources on human rights concerning intersex people that mention the exercise of their rights to self-determination and identity, to be protected from cruel and degrading treatment, and to be effectively included. In addition, there are provisions in Europe and Latin America that prohibit surgeries on intersex children and violence against them.

The analysis of treaties, conventions, regulations, laws, jurisprudence, recommendations, institutional declarations, activist statements and reports evidenced that intersex people in Europe and Latin America experience many forms of discrimination and violence based on stereotypes and myths about what is male and female. Two topics predominate in international human rights law: first, their right to bodily integrity and protection from cruel and degrading treatment in the medical field, and second, their right to self-determination and identity, which is reflected in the possibility of choosing a gender, and these are evidenced in legal documents. These rights have also emerged in United Nations international regulations as a critical element in guaranteeing intersex people's human rights. The YP+10 are essential in this matter and their content appears implicitly or explicitly in the legal developments of European and Latin American countries.

In addition, this research provided insights into how intersex organisations perceive current developments of the human rights framework to protect their rights. For instance, in Europe, intersex organisations have clearly identified violations of the right to integrity and claim for intersex people the right not to be subjected to medical treatments or surgeries during infancy. In this case, within Europe, some states' progress is needed to establish additional mechanisms that will effectively implement the existing human rights frameworks. In contrast, in Latin America, intersex organisations also identify challenges in the general development and recognition of a human rights framework for intersex people. Legal initiatives are present in some Latin American countries to guarantee the rights of intersex people, mainly their rights to bodily integrity, self-determination and identity. However, legal advancements are required concerning the prohibition of sex-normalising surgeries. Another important finding in both regions is the lack of reparation and judicial mechanisms for intersex people who have been the victims of massive human rights violations during their lives. Legal tools that not only acknowledge but also aim to repair their suffering are essential to promote real change. The insights provided by this research reveal how important it is to continue evaluating the legal framework from a comparative perspective to better understand what happens to intersex people in each country within the same region and between regions.

Whether social beliefs regarding gender, and, thus, the political approaches that prevail in both Europe and Latin America, are obstacles to implementing public policies for intersex people remains a pending research question. If European and Latin American societies do not develop a cultural openness toward gender and bodily diversities, intersex individuals will continue to experience violence against their bodies. Consequently, in Europe, it is important to continue working on the effective implementation of regulations to protect the intersex community and the typification of tangible sanctions for non-compliance. In addition, in Latin America, it is essential that states commit to developing policies based on a human rights approach to enable intersex people to have a dignified life and independent recognition. Finally, it is necessary to remember that human rights are not only a tool to better interpret the law and create more efficient public policies, but primarily a means to achieve social justice.

**Funding:** The research forms part of the project INIA. Intersex—New Interdisciplinary Appproaches. This project has received funding from the European Union's Horizon 2020 Research and Innovation Programme under the Marie Skłodowska-Curie grant agreement No. 859869. This paper reflects only the views of the author and the Agency is not responsible for any use that may be made of the information it contains.

**Institutional Review Board Statement:** Not Applicable.

**Informed Consent Statement:** Not Applicable.

**Data Availability Statement:** Not Applicable.

**Conflicts of Interest:** The author declares no conflict of interest.

## Notes

[1]    The European Court of Human Rights. 1999. Case V. vs. the United Kingdom, p. 71.

[2]    The European Court of Human Rights. 1978. Case Irland v. the U.K, pp. 161–62.

[3]    The European Court of Human Rights. 2006. Case Mubilanzila Mayeka and Kaniki Mitunga v. Belgium, p. 17.

[4]    The European Court of Human Rights. 2006b. Case Jalloh v. Germany.

[5]    The European Court of Human Rights. 2001. Case Z vs. the United Kingdom, p. 109.

[6]    Malta Gender Identity, Gender Expression and Sex Characteristics Act No. XI.

[7]    Malta Gender Identity, Gender Expression and Sex Characteristics Act No. XI 2015. Article 17.

[8]    Lei n.º 38/2018. Direito à autodeterminação da identidade de género e expressão de género e à proteção das características sexuais de cada pessoa.

[9]    Lei n.º 38/2018. Direito à autodeterminação da identidade de género e expressão de género e à proteção das características sexuais de cada pessoa. Article 15.

[10]    Iceland Act on Gender Autonomy No 80 /2019 as amended by Act No. 159/2019, No. 152/2020 and No. 154/2020.

[11]    Gesetz zur Änderung personenstandsrechtlicher Vorschriften, May 2013.

[12]    Gesetz zur Änderung der in das Geburtenregister einzutragenden Angaben, December 2018.

[13]    Gesetz zum Schutz von Kindern mit Varianten der Geschlechtsentwicklung, 21 May 2021.

[14]    Chambre des représentants de Belgique, Résolution 0043/008, 11 February 2021.

[15]    Constitución Española de 1978, Título VIII. De la Organización Territorial del Estado, Capítulo tercero. De las Comunidades Autónomas.

[16]    Ley 2/2016, de 29 de marzo, de Identidad y Expresión de Género e Igualdad Social y no Discriminación de la Comunidad de Madrid, Ley 8/2016, de 27 de mayo, de igualdad social de lesbianas, gais, bisexuales, transexuales, transgénero e intersexuales, y de políticas públicas contra la discriminación por orientación sexual e identidad de género en la Comunidad Autónoma de la Región de Murcia, Ley 4/2018, de 19 de abril, de Identidad y Expresión de Género e Igualdad Social y no Discriminación de la Comunidad Autónoma de Aragón, Ley 2/2021, de 7 de junio, de igualdad social y no discriminación por razón de identidad de género, expresión de género y características sexuales en la Comunidad Autónoma de Canarias.

[17]    Anteproyecto de Ley para la igualdad real y efectiva de las personas trans y para la garantía de los derechos de las personas LGTBI, June 2021.

[18]    Anteproyecto de Ley para la igualdad real y efectiva de las personas trans y para la garantía de los derechos de las personas LGTBI, June 2021, Articulo 18.

[19]    Anteproyecto de Ley para la igualdad real y efectiva de las personas trans y para la garantía de los derechos de las personas LGTBI, June 2021, Articulo 71.

[20]    See the Reference Center for Rare Diseases: https://www.developpement-genital.org/ (accessed on 14 June 2022).

[21]    See the rulings of the Colombian Constitutional Court N. T-594 of 1993, T-539 of 1994, T-097 of 1994, SU-623 of 2001, C-577 of 2011, SU-617 of 2014.

[22]    Colombian Constitutional Court, Ruling N. T-477/95, 31–32.

[23]    Colombian Constitutional Court, Ruling N. SU-337/99, 36.

[24]    Colombian Constitutional Cort, Ruling N. SU-337/99, 99–100

[25]    Ley Argentina de Identidad de Género No. 26.743, 24 May.

[26]    Proyecto de Ley sobre la Protección Integral de las Características Sexuales, S-2090/19, November 2020.

[27]    See note 26 above.

[28]    Own translation from Spanish ["la intervención a través de terapias hormonales innecesarias y no consentidas que en repetidas ocasiones han derivado en graves secuelas a la salud del individuo intersexual, pasando por cirugías que se pueden comparar con procedimientos de esterilización forzada y mutilación genital"].

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
