# Peer review of "The Human Rights Situation of Intersex People: An Analysis of Europe and Latin America"

_socsci, doi:10.3390/socsci11070317_

Round 1

Author Response

Dear Reviewers,

Thank you for your very helpful comments and suggestions. I particularly appreciate R1’s feedback regarding the unbalances between the two regions studied and, and R2’s useful contribution concerning the differentiation between sex, gender, and sex characteristics concepts and interpretations.

I have made substantial revisions to the draft paper and hope that it will now be suitable for publication. The suggested revisions (in bold) and responses to comments are as follows:

  1. (R1) The introduction needs to be reshaped so as to become more concise and precise. I reshaped the introduction following a more concise and precise structure.

What is the main issue that needs to be further discussed among activism and legal reform? I thoroughly explained intersex people’s main issues and violations of human rights that need to be discussed.

What is the gap in literature on human rights law that has been left unaddressed so far in relation to intersex rights? I also explained how the main gap is regarding the rule of law in the different legislations in comparison with the international and regional mandates.

Why is it relevant to concentrate on Latin America and Europe? The introduction names the reasons why the two were chosen.

In this regard, it is necessary to better explain the link between discrimination, marginalisation and violence. The three terms appear at the very outset of the paper, but their connection, at least for the purposes of the concerned investigation, is not clear. The link between discrimination, marginalization and violence was removed to make more space to deep into the debate on sexual orientation, gender identity, and sex characteristics.   

  • The author presents ‘theory, practice and activism’ (p. 2) as interlinked. Yet, the manuscript remains silent, at that stage, on the relevance of these three factors for the improvements of intersex lives. Particularly, it is unclear whether the paper is concerned more about law as a tool of social change than about social mobilisation for intersex rights. The former impression comes from the first two pages of the introduction, where the focus seems to be human rights law. Then, the analysis moves to advocacy for intersex rights. This has been addressed by removing the sentence “theory, practice and activism”, as this issue is not a priority within the paper's objectives. The paper was modified to give a more precise focus on human rights law and the protection of intersex people’s rights throughout the paper.
  • Intersex issues are debated now in the field of human rights law.1 I would reconsider the statement ‘little has been said about the role of law in making their lives better’ (p. 1), which dates back to an authoritative but ten-year-old source. This statement also contradicts the subsequent reference to violations upon intersex children as a form of cruel and inhuman treatment – a clear reference to the vocabulary of international human rights law. The mentioned statement was removed so as to not mislead the reader.

  1. (R1) The methodology concentrates on the legal framework and activists’ documents. I suggest the author reconsiders the definition of ‘legal research’ at page 3, so that it reflects the purpose of the paper. The type of legal research on which the manuscript relies is positivistic in that it looks at the law as it is, trying to interpret the protections it enshrines. But the spirit of the (legal) research is also normative, asking what national law should do to enhance the protections against intersex-targeting violations in compliance with international standards. This combined approach could be presented already at the beginning of the article, so that the reader is prepared to the spirit of the investigation since the very outset. The structure of the methodology was changed to clarify the scoping review and legal research. The methodology now is divided into three main parts; the second one incorporates a new definition of “legal research” and better legal interpretation and the introduction names the methodology as well as the combined approach of international standards and national legislation. 

  • The distinction between so-called primary and secondary sources is misleading, since the same group ends up including sources of law that are different in nature, legal force and applicability, for example a judgment and an institutional statement/report. This taxonomy does not mirror the analogies between, and real impact of legal documents. I suggest reconsidering it. While it is true that there are many ways to classify the sources of law, solid criteria should justify the division between ‘primary’ and ‘secondary.’ I have resolved this misleading by removing the categorization of primary and secondary sources. Instead, I have focused on sources without classifying them and explained in more detail the purpose of legal research which included the reviewing of activist documents.

  1. (R1) The discussion of the violations (Section 3.1) would greatly benefit from a description of what happens on the ground, for instance relying on activist organisations’ reports, so that to provide a concrete picture of the facts, and the importance of the protections (particularly, the YP+10). Here, I also recommend that the author explains, before using it, the expression ‘sex normalisation surgeries’ to avoid (albeit unintended) linguistic stigmatisation (p. 7). In the whole section, it remains unclear why gender expression, cited in the title of the section, is relevant, as well as the relation between gender identity and intersex. The question is at the heart of lively debates among activists, arguing, among others, that intersexuality is not necessarily part of the queer universe, that intersex is not a gender identity, that adding a third (fourth, fifth..) gender to IDs is not a solution for stopping surgeries upon intersex children, that the priority is the protection of the right to physical and mental integrity, and many others. Such debates cannot be overlooked, but, at least, referenced. The author should also make clear the difference between gender identity, gender expression and sex characteristics not to run the risk of confusing them while discussing the human rights violations. This clarification would also help understand the reference to gender recognition and trans people that follows in the subsequent section. The purpose of section 3.1 is to provide an overview of the general legal framework of intersex people´s rights to further explain what is happening on the ground in section 3.2. In section (3.2), information about intersex people´s situation was added using reports of activist organizations in Europe to make equal content in comparison with Latin America. In addition, to explain ‘sex normalization surgeries’ and substantiate intersex people´s rights and claims, sexual orientation, gender identity, and sex characteristics were explained in the introduction to avoid confusion.   

  1. (R1) Section 3.2.1 should be renamed ‘Europe’ instead of ‘European Union’ because it discusses the framework of both the EU and the Council of Europe (including the European Court of Human Rights). References to ‘EU’ or ‘European Union’ are not always adequate, and should be replaced by ‘Europe’ or ‘European’ - see, for example, the last paragraph of page 9. The acronym for the European Court of Human Rights is ‘ECtHR’, rather than ‘ECHR’ which refers to the European Convention on Human Rights. The statement of the Portuguese that ‘does not allow medical interventions’ could be rephrased in terms of prohibition. The author, indeed, uses the word ‘exception’ immediately after, evoking a regime that has a prohibition as general rule, with some specific and narrow exceptions when the surgery is admitted (‘the law only allows this kind of procedure when the minor has established a gender identity, and it considers as the only exception to the norm the possibility to perform medical treatments or interventions in cases of proven risk for the intersex person’s health’). The same rephrasing applies to line 417 in the paragraph concerning Spain. Where it corresponds, the term European Union was changed to Europe. Also, the acronyms for the European Court of Human Rights (ECtHR) and Council of Europe (CoE) were modified. The Portuguese Law and Spanish Law statements were reviewed and modified.

  1. (R1) The paper needs a thorough linguistic revision. The paper went through a linguistic revision.

  1. (R2) Abstract. The last sentence mentioned: "conservative ideologies that defend the strict separation between masculine and feminine that ostracise bodily diversity." But I don't agree using masculine and feminine since it refers to gender, and intersex refers to sex. It will be between male and female. Within the reshaping of the introduction, the sentence was changed as follows: “There are countries with conservative cultures and restrictive gender recognition legislation that express difficulties in guaranteeing a dignified life for LGBTI communities…”

  1. (R2) Authors mentioned: "Nonetheless, the sexual and gender diversity framework does not always include intersex people’ s experiences because their struggles are often ignored by this framework within queer and LGBTI communities and movements" I think the authors should highlight that intersex is not a sexual orientation, but it is also not a gender identity. I think maybe this is why the framework does not always include intersex. I think this issues should be expanded and clarified. I have clarified sexual orientation, gender identity, and sex characteristics to avoid misrepresentation and clarified the differentiation that must be considered regarding intersex people´s rights.

  1. (R2) Authors mentioned "However, few European Union nations have strictly prohibited medical procedures and treatments against intersex children." can the author mention some examples? Examples of legislation that strictly forbade medical procedures and treatments against intersex children were added in the introduction and further explained in the 3.2.1 section, lines 801 to 868. The case of Iceland was added in this regard.

  1. (R2) Table 1, should include the number of articles and documents founded on each process. After reshaping the structure of the methodology, it was not necessary to use Table 1.

  1. Conclusion: I think the issue of intersex being a "sex identity"? and not an sexual orientation or gender identity should be address in this section. This issue has been clarified within the introduction to offer a more nuanced understanding of intersex issues to the reader. 

Reviewer 2 Report

Summary:

This manuscript aimed to elucidate these questions by exploring the current situation of the intersex human rights in the European Union and Latin America through a methodological strategy that merges a scoping review, an analysis of activists’ documents, and a legal research framework. The main contribution of this article is to highlight the importance of addressing the issues of intersexual individuals from a human right perspective as it has been done in Europe to reduce violations of their rights. One of the strengths of this manuscript is that it includes social movements and communities documentation to understand to the issue.  

Abstract

1.The last sentence mentioned: "conservative ideologies that defend the strict separation between masculine and feminine that ostracise bodily diversity." But I don't agree using masculine and feminine since it refers to gender, and intersex refers to sex. It will be between male and female. 

Introduction: 

1.Authors mentioned: "Nonetheless, the sexual and gender diversity framework does not always include intersex people’ s experiences because their struggles are often ignored by this framework within queer and LGBTI communities and movements" I think the authors should highlight that intersex is not a sexual orientation, but it is also not a gender identity. I think maybe this is why the framework does not always include intersex. I think this issues should be expanded and clarified. 

2. Authors mentioned "However, few European Union nations have strictly prohibited medical procedures and treatments against intersex children." can the author mention some examples? 

3. Authors mentioned: "Whether the conservative views of society regarding gender, and therefore the political approaches that prevail both in the European Union and Latin America, are ..." I don't think intersex is an issue of gender, it is an issue of sex. Psychosocial vs biology. Please does not use gender and sex like synonym. 

4. Table 1, should include the number of articles and documents founded on each process. 

Conclusion

1. I think the issue of intersex being a "sex identity"? and not an sexual orientation or gender identity should be  address in this section.  

Author Response

Dear Reviewers,

Thank you for your very helpful comments and suggestions. I particularly appreciate R1’s feedback regarding the unbalances between the two regions studied and, and R2’s useful contribution concerning the differentiation between sex, gender, and sex characteristics concepts and interpretations.

I have made substantial revisions to the draft paper and hope that it will now be suitable for publication. The suggested revisions (in bold) and responses to comments are as follows:

  1. (R1) The introduction needs to be reshaped so as to become more concise and precise. I reshaped the introduction following a more concise and precise structure.

What is the main issue that needs to be further discussed among activism and legal reform? I thoroughly explained intersex people’s main issues and violations of human rights that need to be discussed.

What is the gap in literature on human rights law that has been left unaddressed so far in relation to intersex rights? I also explained how the main gap is regarding the rule of law in the different legislations in comparison with the international and regional mandates.

Why is it relevant to concentrate on Latin America and Europe? The introduction names the reasons why the two were chosen.

In this regard, it is necessary to better explain the link between discrimination, marginalisation and violence. The three terms appear at the very outset of the paper, but their connection, at least for the purposes of the concerned investigation, is not clear. The link between discrimination, marginalization, and violence was removed to make more space to deep into the debate on sexual orientation, gender identity, and sex characteristics.   

  • The author presents ‘theory, practice and activism’ (p. 2) as interlinked. Yet, the manuscript remains silent, at that stage, on the relevance of these three factors for the improvements of intersex lives. Particularly, it is unclear whether the paper is concerned more about law as a tool of social change than about social mobilisation for intersex rights. The former impression comes from the first two pages of the introduction, where the focus seems to be human rights law. Then, the analysis moves to advocacy for intersex rights. This has been addressed by removing the sentence “theory, practice and activism”, as this issue is not a priority within the paper's objectives. The paper was modified to give a more precise focus on human rights law and the protection of intersex people’s rights throughout the paper.
  • Intersex issues are debated now in the field of human rights law.1 I would reconsider the statement ‘little has been said about the role of law in making their lives better’ (p. 1), which dates back to an authoritative but ten-year-old source. This statement also contradicts the subsequent reference to violations upon intersex children as a form of cruel and inhuman treatment – a clear reference to the vocabulary of international human rights law. The mentioned statement was removed so as to not mislead the reader.

  1. (R1) The methodology concentrates on the legal framework and activists’ documents. I suggest the author reconsiders the definition of ‘legal research’ at page 3, so that it reflects the purpose of the paper. The type of legal research on which the manuscript relies is positivistic in that it looks at the law as it is, trying to interpret the protections it enshrines. But the spirit of the (legal) research is also normative, asking what national law should do to enhance the protections against intersex-targeting violations in compliance with international standards. This combined approach could be presented already at the beginning of the article, so that the reader is prepared to the spirit of the investigation since the very outset. The structure of the methodology was changed to clarify the scoping review and legal research. The methodology now is divided into three main parts; the second one incorporates a new definition of “legal research” and better legal interpretation and the introduction names the methodology as well as the combined approach of international standards and national legislation. 

  • The distinction between so-called primary and secondary sources is misleading, since the same group ends up including sources of law that are different in nature, legal force and applicability, for example a judgment and an institutional statement/report. This taxonomy does not mirror the analogies between, and real impact of legal documents. I suggest reconsidering it. While it is true that there are many ways to classify the sources of law, solid criteria should justify the division between ‘primary’ and ‘secondary.’ I have resolved this misleading by removing the categorization of primary and secondary sources. Instead, I have focused on sources without classifying them and explained in more detail the purpose of legal research which included the reviewing of activist documents.

  1. (R1) The discussion of the violations (Section 3.1) would greatly benefit from a description of what happens on the ground, for instance relying on activist organisations’ reports, so that to provide a concrete picture of the facts, and the importance of the protections (particularly, the YP+10). Here, I also recommend that the author explains, before using it, the expression ‘sex normalisation surgeries’ to avoid (albeit unintended) linguistic stigmatisation (p. 7). In the whole section, it remains unclear why gender expression, cited in the title of the section, is relevant, as well as the relation between gender identity and intersex. The question is at the heart of lively debates among activists, arguing, among others, that intersexuality is not necessarily part of the queer universe, that intersex is not a gender identity, that adding a third (fourth, fifth..) gender to IDs is not a solution for stopping surgeries upon intersex children, that the priority is the protection of the right to physical and mental integrity, and many others. Such debates cannot be overlooked, but, at least, referenced. The author should also make clear the difference between gender identity, gender expression and sex characteristics not to run the risk of confusing them while discussing the human rights violations. This clarification would also help understand the reference to gender recognition and trans people that follows in the subsequent section. The purpose of section 3.1 is to provide an overview of the general legal framework of intersex people´s rights to further explain what is happening on the ground in section 3.2. In section (3.2), information about intersex people´s situation was added using reports of activist organizations in Europe to make equal content in comparison with Latin America. In addition, to explain ‘sex normalization surgeries’ and substantiate intersex people´s rights and claims, sexual orientation, gender identity, and sex characteristics were explained in the introduction to avoid confusion.   

  1. (R1) Section 3.2.1 should be renamed ‘Europe’ instead of ‘European Union’ because it discusses the framework of both the EU and the Council of Europe (including the European Court of Human Rights). References to ‘EU’ or ‘European Union’ are not always adequate, and should be replaced by ‘Europe’ or ‘European’ - see, for example, the last paragraph of page 9. The acronym for the European Court of Human Rights is ‘ECtHR’, rather than ‘ECHR’ which refers to the European Convention on Human Rights. The statement of the Portuguese that ‘does not allow medical interventions’ could be rephrased in terms of prohibition. The author, indeed, uses the word ‘exception’ immediately after, evoking a regime that has a prohibition as general rule, with some specific and narrow exceptions when the surgery is admitted (‘the law only allows this kind of procedure when the minor has established a gender identity, and it considers as the only exception to the norm the possibility to perform medical treatments or interventions in cases of proven risk for the intersex person’s health’). The same rephrasing applies to line 417 in the paragraph concerning Spain. Where it corresponds, the term European Union was changed to Europe. Also, the acronyms for the European Court of Human Rights (ECtHR) and Council of Europe (CoE) were modified. The Portuguese Law and Spanish Law statements were reviewed and modified.

  1. (R1) The paper needs a thorough linguistic revision. The paper went through a linguistic revision.

  1. (R2) Abstract. The last sentence mentioned: "conservative ideologies that defend the strict separation between masculine and feminine that ostracise bodily diversity." But I don't agree using masculine and feminine since it refers to gender, and intersex refers to sex. It will be between male and female. Within the reshaping of the introduction, the sentence was changed as follows: “There are countries with conservative cultures and restrictive gender recognition legislation that express difficulties in guaranteeing a dignified life for LGBTI communities…”

  1. (R2) Authors mentioned: "Nonetheless, the sexual and gender diversity framework does not always include intersex people’ s experiences because their struggles are often ignored by this framework within queer and LGBTI communities and movements" I think the authors should highlight that intersex is not a sexual orientation, but it is also not a gender identity. I think maybe this is why the framework does not always include intersex. I think this issues should be expanded and clarified. I have clarified sexual orientation, gender identity, and sex characteristics to avoid misrepresentation and clarified the differentiation that must be considered regarding intersex people´s rights.

  1. (R2) Authors mentioned "However, few European Union nations have strictly prohibited medical procedures and treatments against intersex children." can the author mention some examples? Examples of legislation that strictly forbade medical procedures and treatments against intersex children were added in the introduction and further explained in the 3.2.1 section, lines 801 to 868. The case of Iceland was added in this regard.

  1. (R2) Table 1, should include the number of articles and documents founded on each process. After reshaping the structure of the methodology, it was not necessary to use Table 1.

  1. Conclusion: I think the issue of intersex being a "sex identity"? and not an sexual orientation or gender identity should be address in this section. This issue has been clarified within the introduction to offer a more nuanced understanding of intersex issues to the reader. 

Round 2

Reviewer 1 Report

The quality of the article has now improved, since the structure meets the established aim of exploring intersex issues from the twofold angle of legal texts and activists' reports. In terms of content, the paper lacks specific solid argumentation in support of crucial statements, as explained in the attached file. However, the English structure and style need a major review. 

Author Response

Dear Reviewer,

Thank you for your very helpful comments and suggestions. I appreciate the exhaustive work made with the correction of the details to give more coherence to the manuscript.

I have made substantial revisions to the draft paper and hope that it will now be suitable for publication. The suggested revisions (in bold) and responses to comments are as follows:

  1. General Comment: The quality of the article has now improved since the structure meets the established aim of exploring intersex issues from the twofold angle of legal texts and activists' reports. In terms of content, the paper lacks specific solid argumentation in support of crucial statements, as explained in the attached file. However, the English structure and style need a major review. The arguments highlighted were better explained to give more coherence to the manuscript. In addition, the paper went through the editing English revision of the Journal. Please see the attachment. 
  1. Line 59: This statement is not supported by any solid argument. Please rephrase so as to make it more nuanced and attributable to activists' voices. The mentioned statement was removed in order not to mislead the reader.
  2. Line 160: This conclusion comes suddenly, without any specific discussion concerning the reasons why Latin America and Europe are compared. Please provide a deeper argumentation supporting this statement. I have explained better this argument and provided references to support it.
  3. There is inconsistency in terms of personal and impersonal voice. Please decide whether to opt for 'I' or keep it impersonal. The personal voice comes out only starting from this Section 2. I have modified the inconsistencies by rewriting the manuscript only with an impersonal voice.
  4. Line 834: I am not sure it is useful to distinguish between the two since the scoping review of legal texts implicitly includes an analysis/research which is legal in nature. I will maintain the distinction throughout the manuscript since the methodology used for the scoping review was different from the one used in the legal research. For the scoping review, I researched specific keywords related to intersex people and human rights in databases that did not provide enough legal material. At the same time, I conducted legal research where I started to analyze each continent's regulations separately, as well as the activist opinions of each legislation. I did the same procedure for the reviewed national legislations. For the legal research, I also took into consideration legislation and regulations of non-discrimination based on sex and gender, different from the scoping review, where the selection criteria were strictly focused on intersex people issues.
